# Methylene Blue Release from Chitosan/Pectin and Chitosan/DNA Blend Hydrogels

**DOI:** 10.3390/pharmaceutics13060842

**Published:** 2021-06-07

**Authors:** Cassiele T. Cesco, Artur J. M. Valente, Alexandre T. Paulino

**Affiliations:** 1Department of Food and Chemical Engineering, Santa Catarina State University, Pinhalzinho 89870-000, Brazil; cassi.taffarel@hotmail.com; 2Department of Chemistry, University of Coimbra, 3004-535 Coimbra, Portugal; 3Postgraduate Program in Applied Chemistry, Santa Catarina State University, Joinville 89219-710, Brazil

**Keywords:** hydrogel, release, methylene blue, DNA, chitosan, pectin

## Abstract

Chitosan/DNA blend hydrogel (CDB) and chitosan/pectin blend hydrogel (CPB) were synthesized using an emulsion (oil-in-water) technique for the release of methylene blue (model molecule). Both hydrogels were characterized by swelling assays, Fourier transform infrared (FT-IR) spectroscopy, thermogravimetric analysis (TGA) and scanning electron microscopy (SEM), before and after the methylene blue (MB) loading. Higher swelling degrees were determined for both hydrogels in simulated gastric fluid. FT-IR spectra inferred absorption peak changes and shifts after MB loading. The TGA results confirmed changes in the polymer network degradation. The SEM images indicated low porosities on the hydrogel surfaces, with deformed structure of the CPB. Smoother and more uniform surfaces were noticed on the CDB chain after MB loading. Higher MB adsorption capacities were determined at lower initial hydrogel masses and higher initial dye concentrations. The MB adsorption mechanisms on the hydrogel networks were described by the monolayer and multilayer formation. The MB release from hydrogels was studied in simulated gastric and intestinal fluids, at 25 °C and 37 °C, with each process taking place at roughly 6 h. Higher release rates were determined in simulated gastric fluid at 25 °C. The release kinetics of MB in chitosan/DNA and chitosan/pectin matrices follows a pseudo-second-order kinetic mechanism.

## 1. Introduction

Bioactive release systems have been widely studied to minimize the side effects of orally administered medicine methods [1,2]. High medicine doses in human organisms normally cause toxicity, whereas low doses can be ineffective [3]. Therefore, release systems are necessary to improve the therapeutic results, reducing the dose quantities and concentrations of medicines applied in target sick cells. Moreover, release processes improve the therapeutic actions of medicines during control, treatment and healing of different diseases [1].

MB is a cationic dye (Figure 1) used in medical areas as it has key microbiology and pharmacology properties for the treatment of diseases, including sepsis and cancer [4,5]. This dye can be essential for the treatment of tumor tissues due to its selective accumulation capacity in damaged cells. Overall, MB can be employed to treat either cancer or other types of diseases when applying electromagnetic irradiation emission techniques [4]. The release of MB in target cells is also useful due to its easy reduction in biological media and excretion from organisms.

There are many types of solid matrices employed as drug carriers for release systems, including hydrogels [6]. These materials are commonly synthesized using natural biopolymers with the formation of hydrophilic three-dimensional polymeric networks capable of absorbing high volumes of either water or biological fluids [7]. Hydrogels are useful in the encapsulation and release of drugs when highly porous structures are formed [8]. Hydrogel porous chemical structures might be responsive to, for instance, ionic strength, pH and temperature, by swelling and thus facilitating the release processes of encapsulated drugs [9,10].

Natural polysaccharides are potential matrices for hydrogel synthesis as they are biocompatible, biodegradable, non-toxic, eco-friendly and inexpensive compounds [11]. Chitosan is one of the most applied polysaccharides in hydrogel synthesis due to its antimicrobial activity, biodegradability and biocompatibility for living beings [12]. Additionally, the interaction of many solutes, such as drugs and dyes, with chitosan has been widely studied over the years [13,14,15,16]. Chitosan is produced from the chitin deacetylation, which is commonly extracted from, e.g., crustacean exoskeletons and silkworm chrysalides [17]. Covalently cross-linked chitosan-based hydrogels are extremely important in drug release systems [18]. However, the final biomaterial shows cytotoxicity when the crosslinking process is carried out with potentially toxic monomers. This problem is overcome by synthesizing physically cross-linked hydrogels [2].

Pectin is a widely produced natural polysaccharide from primary cell walls of citrus peels, which are considered agro-industrial residues. Hydrogels can be prepared by the coacervation technique using mixtures of polysaccharides such as pectin and chitosan [11] and mixtures of polysaccharides and deoxyribonucleic acid (DNA). DNA is a biopolymer containing deoxyribose sugar capable of carrying genetic information, which is useful in hydrogel synthesis. DNA-based hydrogels have unique properties such as biocompatibility, selective connection, and molecular recognition [19,20,21]. This enables their applications in drug-released systems and tissue engineering [2].

The aim of this work was to prepare and characterize the coacervate chitosan/DNA blend and chitosan/pectin blend hydrogels for the loading and release of methylene blue (model molecule). These hydrogels were obtained using an emulsion (oil-in-water) technique and characterized by swelling assays, FT-IR, TGA and SEM. The release kinetics of MB to different media were assessed and discussed by using different kinetic models.

## 2. Material and Methods

### 2.1. Reagents

Chitosan (CS, 50,000 to 190,000 Da) with 76% deacetylation degree, deoxyribonucleic acid (DNA, sodium salt from salmon testes, 20 kDa) and pectin from citrus peel (PC, *M*_w_ = 9000) with galacturonic acid ≥74.0%, were purchased from Sigma-Aldrich^®^. Methylene blue (MB), dipotassium phosphate (K_2_HPO_4_), acetic acid (CH_3_COOH), phosphoric acid (H_3_PO_4_), hydrated sodium acetate (C_2_H_3_NaO_2_.3H_2_O), hydrochloric acid (HCl) and sodium hydroxide (NaOH) were purchased from Honeywell^®^ Company. Potassium dihydrogen phosphate (KH_2_PO_4_) was purchased from M&B^®^ Company. All reagents used were of analytical grade and all solutions were prepared with Milli-Q ultrapure water.

### 2.2. Hydrogel Synthesis

The hydrogel syntheses were performed by coacervation using an emulsion (oil-in-water) technique described by Filho et al. [22]. The chitosan/DNA hydrogel was initially obtained by preparing a 1% (*w*/*v*) chitosan aqueous solution in acetate buffer solution at pH 3.0 and 1% (*w*/*v*) salmon DNA aqueous solution in phosphate buffer solution at pH 6.0, stirring both solutions for 12 h at room temperature. Next, 1.0 mL of chitosan and 1.0 mL of DNA solutions were mixed with 5.0 mL of benzyl alcohol and homogenized using an Ultra-Turrax (X 1000D Unidrive-Cat-Ing), at 14,000 rpm for 2 min. The chitosan/pectin hydrogel was prepared using a 1% (*w*/*v*) pectin aqueous solution prepared in phosphate buffer solution at pH 9.2. In this case, 1.0 mL of chitosan solution was mixed with 1.0 mL of pectin solution in a flask containing 5.0 mL of benzyl alcohol. This system was also homogenized as described before. The chitosan/DNA and chitosan/pectin emulsions formed during the homogenization processes were filtered using filter paper to remove insoluble compounds and washed five times with acetone to remove unreacted compounds. Hydrogels were frozen in an ultra-freezer (KUNFT Kuf2532 Wh) and freeze-dried (Free Zone 4.5-Labconco) at −55.0 ± 1.0 °C for 24 h [22] prior to MB loading and release experiments taking place.

### 2.3. Swelling Kinetic

Dried hydrogel samples with approximately 70.0 mg were placed in beakers containing 50.0 mL of either ultrapure water, simulated physiological fluid (pH 7.01), simulated gastric fluid (pH 1.21) or simulated intestinal fluid (pH 6.52). The masses of swollen hydrogels were measured at different times (from 3 to 2880 min). The degree of swelling (*DS*) was calculated using Equation (1):(1)$$DS=\frac{w_{s}\left(t\right)-w_{d}}{w_{d}}$$
in which *w_s_(t)* is the weight (g) of the swollen gel, at a specific swelling time, and *w_d_* is the dried gel weight (g).

The swelling mechanism was studied using the power law equation [23] (Equation (2)):(2)$$\frac{w_{t}}{w_{eq}}=kt^{n}$$
in which *w_t_* and *w_eq_* are the masses (g) of the absorbed water by the gel network at specific absorption time and equilibrium, respectively, *n* is the exponent describing the water diffusion mechanism, *k* is the constant, and *t* is the swelling time (min).

The water diffusion parameters were determined by considering the swelling of 60% of the biopolymer network due to the linear relationship between log(*w*_t_/*w*_eq_) and log(*t*) [24]. Thus, the linear form of Equation (2) gives (Equation (3)):(3)$$\log\left(\frac{w_{t}}{w_{eq}}\right)=\log\left(k\right)+n\log\left(t\right)$$

### 2.4. Characterization of Hydrogels

#### 2.4.1. Sample Preparation for Characterization

The characterization studies of hydrogels were performed before and after the MB loading. Gel samples without adsorbed MB were immersed in ultrapure water for 24 h until achieving the swelling equilibrium. Next, these samples were frozen in liquid nitrogen, fragmented, and freeze-dried (Free Zone 4.5-Labconco) at −55.0 ± 1.0 °C for 24 h prior to characterization. MB was loaded into hydrogel network by immersing dried material samples in 10.0 mg L^−1^ MB solution under 110 rpm constant stirring and at 25.0 ± 1.0 °C, for 24 h.

#### 2.4.2. Fourier-Transform Infrared (FT-IR) Spectroscopy

Fourier-transform infrared spectra were recorded using an Agilent Cary 630 FTIR spectrometer with attenuated total reflectance (ATR), at a spectral range from 4000 to 500 cm^−1^, and 32 scans per sample.

#### 2.4.3. Thermogravimetric Analyzes (TGA)

Thermogravimetric analyzes were performed using a TG209 F3 Tausus Netzsh operating from 25 to 600 °C, with a heating rate of 10 °C min^−1^, and under N_2_ atmosphere with flow of 50 mL min^−1^.

#### 2.4.4. Scanning Electron Microscopy (SEM)

Previously prepared hydrogel samples were covered with a thin gold film to increase the electrical conductivity of the material surface prior to the measurements by scanning electron microscopy (Tescan-Vegas 3).

### 2.5. Loading of MB by Adsorption

The MB adsorption/loading process in the chitosan/DNA hydrogel (CDB) and chitosan/pectin hydrogel (CPB) was conducted by immersing dried hydrogel pieces in Erlenmeyers containing MB aqueous solutions. Next, these flasks were placed on a shaker incubator (Labwit ZWY-103B) operating at 25.0 ± 1.0 °C for 24 h. The effect of initial hydrogel weight was evaluated by using polymer samples of 5 to 30 mg. On the other hand, the effect of the initial MB concentrations was assessed in the concentration range: 0.05 to 15 mg L^−1^. For all experiments, a volume of MB solution equal to 5.0 mL and a stirring speed of 90 rpm were used. The remaining MB concentrations in solutions were determined by UV-Vis spectrophotometry (Shimadzu-UV-2450) at 664 nm. The MB adsorption capacities (*q_e_*) of the hydrogels were computed using Equation (4):(4)$$q_{e}=\frac{\left(C_{0}-C_{eq}\right)}{m}*\;V$$
in which *C_0_* (mg L^−1^) is the initial MB concentration, *C_eq_* (mg L^−1^) is the equilibrium concentration, *m* (g) is the dried hydrogel weight, and *V* (L) is the initial volume of aqueous solution [25].

The MB encapsulation (*EE*) and loading (*LE*) efficiencies were, respectively, determined using Equations (5) and (6):(5)$$EE=\frac{\left(total\;MB\;mass-free\;MB\;mass\right)}{total\;MB\;mass}*\;100$$
(6)$$LE=\frac{\left(total\;MB\;mass-free\;MB\;mass\right)}{biohydrogel\;mass}*\;100$$

### 2.6. Adsorption Isotherm Models

#### 2.6.1. Langmuir Isotherm

The two-parameter Langmuir isotherm model is employed to describe the adsorption mechanism occurring with monolayer formation of adsorbates on adsorbent surfaces [26]. The two-parameter Langmuir isotherm mathematical model is represented by Equation (7):(7)$$q_{e}=\frac{q_{max}K_{L}C_{e}}{1+K_{L}C_{e}}$$
in which *q_max_* is the maximum Langmuir adsorption capacity (mg g^−1^), *K_L_* is the Langmuir constant related to adsorption rate (L mg^−1^), and *C_e_* is the equilibrium adsorbate concentration (mg L^−1^) [26].

The nature and feasibility of the adsorption process can be assessed by the quantification of the Langmuir separation factor (*R_L_*), as described by Equation (8):(8)$$R_{L}=\frac{1}{1+K_{L}C_{e}}$$

#### 2.6.2. Freundlich Isotherm

The two-parameter Freundlich isotherm model is obtained from information of the two-parameter Langmuir isotherm model. This model is employed to describe the multilayer formation and interactions occurring among chemical species adsorbed on/in adjacent active sites in the adsorbent structure [27] and is described by Equation (9):(9)$$q_{e}=K_{F}C_{e}^{\frac{1}{n}}$$
in which *K_F_* (mg^(*n*−1)/*n*^ L^1/*n*^g^−1^) is the Freundlich constant related to adsorption capacity, and *1/n* is the Freundlich constant related to adsorbent surface heterogeneity [27].

#### 2.6.3. Redlich-Peterson Isotherm

The three-parameter Redlich–Peterson isotherm model is described by combining parameters of the two-parameter Langmuir and Freundlich isotherms [28] (Equation (10)):(10)$$q_{e}=\frac{K_{R-P}C_{e}}{1+a_{R-P}C_{e}^{\beta_{R-P}}}$$
in which *α_R−P_* (mg^−1^) and *K_R−P_* (L g^−1^) are Redlich–Peterson constants, and *β*_R*−*P_ is the Redlich–Peterson exponent [28].

#### 2.6.4. Sips Isotherm

The three-parameter Sips isotherm model is also described by using information of the two-parameter Langmuir and Freundlich isotherms [29] as it comes from Equation (11):(11)$$q_{e}=\frac{q_{ms}K_{s}C_{e}^{\beta_{S}}}{1+K_{S}C_{e}^{\beta_{S}}}$$
in which *q_ms_* (mg g^−1^) is the maximum Sips adsorption capacity, *K_s_* (L mg^−1^) is the Sips equilibrium constant, and *β_s_* is the Sips exponent employed to explain the homogeneity/heterogeneity of the adsorption system [29].

### 2.7. MB Release Kinetics

Chitosan/DNA and chitosan/pectin hydrogel samples of approximately 10.0 mg containing MB were immersed in 100.0 mL of either simulated gastric fluid (SGF) or simulated intestinal fluid (SIF). Then, the flasks were placed on a shaker incubator (Labwit ZWY-103B) under 110 rpm constant stirring at 25.0 ± 1.0 or 37.0 ± 1.0 °C. Aliquots of aqueous solutions (2.0 mL) were collected at different times for the quantification of the MB released. That was carried out by UV-Vis spectrophotometry (Shimadzu UV-2450) at 664 nm. The release kinetics were evaluated using the pseudo-first- and pseudo-second-order kinetic models represented by Equations (12) and (13), respectively:(12)$$q_{t}=q_{e}\left(1-e^{-tk_{1}}\right)$$
(13)$$q_{t}=\frac{q_{e}^{2}k_{2}t}{q_{e}k_{2}t+1}$$
in which *k_1_* and *k_2_* are the first- and second-order rate constants, respectively, and *q_t_* is the cumulative release at time *t* [30].

### 2.8. Error Analysis

The correlation coefficient (R²), chi-square statistic test (*χ*²) and Akaike information criterion (AIC) were employed to assess the best fitting procedure. The R^2^, *χ*² and AIC values were determined using Equations (14–16), respectively:(14)$$R^{2}=\frac{\sum {\left(q_{e,p}-\bar{q_{e}}\right)}^{2}}{\sum {\left(q_{e,p}-\bar{q_{e}}\right)}^{2}+\sum {\left(q_{e,p}-{\;q}_{e}\right)}^{2}}$$
(15)$$\chi^{2}=\sum \frac{{\left(q_{e}-q_{e,p}\right)}^{2}}{q_{e,p}}$$
(16)$$AIC=n\log\left(\frac{s^{2}}{n}\right)+2K$$
in which *q_e_* (mg g^−1^) is the adsorption capacity determined from experiments, *q_e,p_* (mg g^−1^) is the adsorption capacity obtained from predicted data, $\bar{q_{e}}$ is the average of *q_e_*, *s*^2^ is the residual sum of squares, *n* is the number of experimental replicates, and *K* is the number of model parameters [23,31,32]. 

## 3. Results and Discussion

### 3.1. Swelling Kinetics

Figure 2 shows the degrees of swelling for the chitosan/pectin (a) and chitosan/DNA (b) blend hydrogels in different media.

The degree of swelling of the chitosan/pectin hydrogel in ultrapure water was approximately 9.59 g of water per g of dried chitosan/pectin gel after 1440 min. This value corresponds to 6.27 g of water per g of dried chitosan/DNA. The swelling degrees for these hydrogels in simulated physiological fluid were, respectively, 4.88 and 2.31 g g^−1^. In simulated intestinal fluid, the DS for CPB and CDB were: 2.559 and 4.254 g g^−1^, respectively. The swelling degrees in simulated gastric fluid were much higher than those found in the other media: 20.83 and 12.88 g g^−1^ for CPB and CDB, respectively. This might be justified as being due to protonation of chitosan amino groups and pectin carboxylic groups in more acidic aqueous solutions. Cationic groups generate electrostatic repulsion forces expanding the polymer network, favoring the diffusion process through pores [33]. This phenomenon exposes polar groups interacting with water due to the solvation process. Moreover, the presence of H^+^ ions in more acidic solutions favors the formation of hydrogen bonds between hydrogel and water, increasing the swelling capacity. Overall, the swelling mechanism of hydrophilic three-dimensional polymeric networks depends on the water diffusion process and macromolecular relaxation of polymer networks [34].

On the other hand, the lowest degrees of swelling were determined for chitosan/pectin hydrogel in intestinal fluid and chitosan/DNA in physiological fluid. Such behavior can be related with the influence of anionic species from salts employed to prepare the aqueous solutions. These species can interact with cationic amine groups in the biopolymer networks, decreasing the electrostatic repulsion forces and swelling capacity. The increase in pH can increase the degree of swelling due to deprotonation of amino and carboxylic groups in the polymeric networks. In this case, the predominant electrostatic repulsion forces take place among anionic carboxylic groups to expand the polymeric network. 

Overall, the swelling mechanism in acidic media mainly takes place due to electrostatic repulsion forces among cationic amine groups, whereas in alkaline media, it mainly takes place due to electrostatic repulsion forces among anionic carboxyl groups. Moreover, intermolecular interactions among amine/carboxylic groups and water molecules favor the swelling processes [35].

Figure 3 shows the water diffusion rates (a and b) and linear regressions for 60% of sorbed water (c and d) in chitosan/pectin (a and c) and chitosan/DNA (b and d) blend hydrogels. The swelling parameters and error analyses are shown in Table 1.

The *n* values during the swelling of the chitosan/pectin and chitosan/DNA hydrogels ranged from 0.081 to 0.401, inferring that the water transport through the polymer networks takes place by diffusion processes (pseudo-Fickian process) without the significant occurrence of relaxation macromolecular. These results were confirmed by the higher R^2^ values and lower χ² and AIC values. The water rate constant (*k*) values were higher during the swelling of the chitosan/pectin hydrogel in the four study media, indicating higher water absorption rates [36]. This is in agreement with the time needed to attain the water absorption equilibrium for both hydrogels. 

### 3.2. Characterization of Hydrogels

#### 3.2.1. Fourier-Transform Infrared (FT-IR) Spectra

Figure 4 shows FT-IR spectra for chitosan/pectin (Figure 4a) and chitosan/DNA (Figure 4b) hydrogels without (CPB and CDB, respectively) and with loaded MB (CPBM and CDBM, respectively).

Absorption bands appearing at roughly 3400 cm^−1^ are attributed to the stretching of O-H and N-H groups from polysaccharide molecules [37]. The absorption bands between 2935 and 2950 cm^−1^ correspond to the either symmetrical or asymmetric stretching of C-H groups from aliphatic structures in biopolymer networks. The absorption bands close to 1634 cm^−1^ are also associated to the N-H folding, C-N stretching, and C=O bond stretching from carboxylate anions [38]. The C=O stretching from amide, carboxylic or ester groups present in pectin/chitosan macromolecules were noticed at 1742 cm^−1^. This absorption band disappeared in the chitosan/DNA hydrogel spectrum due to absence of pectin in the biomaterial structure and intermolecular interactions between chitosan and DNA. The absorption band at 1420 cm^−1^ corresponds to C-H group symmetrical deformation due to the presence of saturated carbon atoms in the polysaccharide molecular structures. Absorption bands due to the C-O elongation and O-H vibration were observed at 1065 and 1220 cm^−1^, respectively [37]. Some absorption bands were practically similar after loading MB in the hydrogel networks, whereas others decreased (C=O and N-H) and shifted (O-H and N-H) when compared with their original appearances. Finally, new bands were found due to intermolecular interactions between dye and hydrogel active functional groups.

#### 3.2.2. Thermogravimetric Analysis

Figure 5 shows thermogravimetric curves (a and c) and their derivatives (b and d) for both blends before and after the MB encapsulation. The thermal analysis parameters are shown in Table 2.

The first thermal degradation evidence was noticed from 60 to 100 °C in all thermograms due to water weight loss. Even after the drying processes in the ovens, adsorbed non-freezing [39] water molecules may still exist. The second thermal degradation profile was noticed from 200 to 400 °C, with decomposition peaks of polymeric monomers at ≈220 °C (derivative curves). In this case, the total weight loss ranged from 12.62 to 21.04%. The maximum thermal degradation temperature (*T*_max_) was higher for CDBM as empty volumes available in hydrogel networks are occupied by solute molecules, hindering the water diffusion though pores [5]. Lower total weight losses were determined for the CPBM and CDBM samples as there are strong intermolecular interactions between MB and hydrogel during the active site occupation of the biomaterial networks, corroborating the SEM results presented below.

#### 3.2.3. Scanning Electron Microscopy (SEM)

Figure 6 shows the images of scanning electron microscopy for the chitosan/pectin (a and b) and chitosan/DNA (c and d) blend hydrogels before loading MB (a and c) and after loading MB (b and d).

A more dense and non-porous surface was noticed for the chitosan/pectin hydrogel, whereas a more porous surface was noticed for the chitosan/DNA hydrogel. More dense structures are generally associated to higher mechanical resistances [2]. A leaf and deformed structure were noticed on the chitosan/pectin hydrogel surface after loading MB, with the appearance of interconnected pores, whereas a smoother surface was noticed for the chitosan/DNA hydrogel after loading MB. Smoother surfaces can indicate that the interaction of MB with DNA is stronger than that with pectin. This corroborates the higher thermal degradation temperature and lower thermal degradation percentage for CDBM (Table 2). Hydrogel biodegradation processes are also evidenced by reversible chemical interactions between polymer monomers and dyes [2], and the porosity of chemically cross-linked hydrogel networks [35,36]. This is clear when comparing physically and chemically cross-linked polymeric networks. The chitosan/DNA hydrogel structure became noticeably less rough after loading MB, indicating that dye molecules can occupy the polymer network pores and interact with hydrogel active sites [2].

### 3.3. Methylene Blue (MB) Adsorption Assays

#### 3.3.1. Effects of Dried Hydrogel Mass and Initial MB Concentration

Overall, the MB encapsulation and loading efficiencies ranged, respectively, from 62.1 to 85.2% and 0.12 to 0.51% by varying the initial chitosan/pectin hydrogel mass from approximately 6.0 to 31.0 mg. The MB encapsulation and loading efficiencies ranged, respectively, from 34.5 to 68.6% and 0.09 to 0.27% for the chitosan/DNA hydrogel. The MB encapsulation and loading efficiencies in the chitosan/pectin hydrogel ranged, respectively, from 43.7 to 56.7% and 0.03 to 0.54% by ranging the initial MB concentrations from 1.0 to 20.0 mg L^−1^. These values ranged, respectively, from 11.5 to 37.3% and 0.008 to 0.24% for the chitosan/DNA hydrogel.

Figure 7 shows the effects of dried hydrogel mass (a) and initial MB concentration (b) in the MB adsorption capacities of the chitosan/pectin blend hydrogel (CPB) and chitosan/DNA blend hydrogel (CDB).

The MB adsorption capacities decreased with the increase of the hydrogel mass, inferring that the prepared biomaterials first absorb water, and subsequently dye. Therefore, the dye adsorption phenomenon takes place by partition, and depends on the water diffusion mechanism [25]. The highest loading values are equal to 5.1 and 3.1 mg g^−1^, for CDB and CPB, respectively, in agreement with the TG and SEM analysis. However, at 10.0 mg, the loading capacity values are still acceptable. Posterior experiments were performed with 10.0 mg of dried hydrogel to minimize errors and evaluate the possible scale increase. The increase in the initial MB concentration increased the adsorption capacity due to higher amount of dye diffused into the biopolymeric network. These values started to stabilize from 17.5 and 20.0 mg L^−1^ during adsorption to the chitosan/DNA and chitosan/pectin hydrogel, respectively, due to saturation of the active adsorption sites in the hydrophilic three-dimensional structures. Overall, the initial MB concentrations ranging from 17.5 to 20.0 mg L^−1^ are suitable for MB loading in both hydrogels. 

#### 3.3.2. Adsorption Isotherms

Figure 8 shows the results of nonlinear Redlich–Peterson and Sips isotherm models for the adsorption of MB to the chitosan/pectin (a) and chitosan/DNA (b) blend hydrogels. Other isotherm model equations, such as Langmuir and Freundlich equations, were used without success to evaluate alternative mechanisms (data not shown). The isotherm parameters are summarized in Table 3.

The isotherms that best fit MB adsorption data onto chitosan/pectin and chitosan/DNA hydrogels were the Redlich–Peterson and Sips models, respectively. This conclusion was based on the higher R² values, in addition to the lower χ² and AIC values. Once the *β*_R-P_ value of the Redlich–Peterson isotherm was similar to 1 (with *α*_R-P_ → *K*_L_ of Langmuir) and the *K*_s_ value of the Sips isotherm was practically zero, the MB adsorption process to the chitosan/pectin hydrogel network tended to take place with both mono- and multilayers. Higher *α*_R-P_ → *K*_L_ values indicate higher adsorption rates and energies during adsorbate layer formations in/on polymer structures [40]. It can also explain the higher adsorption capacity of the chitosan/pectin hydrogel when comparing with the chitosan/DNA hydrogel. On the other hand, adsorption of MB to the chitosan/DNA hydrogel was mostly governed by monolayer formation as the Sips isotherm model approaches to the Langmuir isotherm model at high initial adsorbate concentrations [29]. The maximum adsorption capacity tends to decrease when adsorption processes take place only with monolayer formation due to faster saturation of active adsorption sites on adsorbent surfaces. Overall, Langmuir parameters can also be used for understanding the MB adsorption mechanism in both matrices. The maximum adsorption capacities of the chitosan/pectin and chitosan/DNA hydrogels were, respectively, 116.9 and 11.6 mg of MB per g of xerogel, according to the Langmuir isotherm model. The Langmuir R_L_ values ranged from 0 and 1, inferring that the adsorption processes are, in both cases, somehow favorable [36]. However, the K_L_ value was much lower for the adsorption of MB onto chitosan/pectin hydrogel than that found for chitosan/DNA hydrogel. This indicates that the intermolecular interactions and chemical affinity between adsorbent and adsorbate tend to be much more significant when using MB and chitosan/DNA hydrogel [27]. This is in agreement with the results of thermal analysis and scanning electron microscopy. Therefore, the higher adsorption capacity of the chitosan/pectin hydrogel is probably related with the monolayer and multilayer formation simultaneously during the adsorption process, occupying higher amounts of active adsorption sites in/on the macromolecular structure.

### 3.4. MB Release

The contents of MB released from the chitosan/pectin hydrogel in gastric fluid were 55.1 and 49.2% at 25.0 ± 1.0 and 37.0 ± 1.0 °C, respectively, after 6h. These values were 96.4 and 88.0% from the chitosan/DNA hydrogel, respectively. The contents of MB released from the chitosan/pectin hydrogel in intestinal fluid were 44.0 and 33.5%, at 25.0 ± 1.0 and 37.0 ± 1.0 °C, respectively, after 6 h, whereas from the chitosan/DNA gel were, respectively, 47.4 and 31.4%. However, in simulated gastric fluid, the MB release process is more significant; two different processes might justify this behavior. The protonation of amine groups (-NH_3_^+^) in the chitosan and DNA increase the intermolecular electrostatic repulsion and, consequently, an expansion of polymer networks occurs, increasing the degree of swelling and solute release capacity [41]; in a similar way, the swelling might occur as a consequence of the screening and salting-in effects leading to a polymer network expansion [42]. The solute release capacity is also affected by the exposition and solvation of three-dimensional polymer network hydrophilic groups [43]. The MB release rates from hydrogels in simulated physiological fluid were not significant due to weak electrostatic repulsion forces and the low degree of swelling of the polymer networks (results not shown). Overall, the chitosan/DNA blend demonstrated to be a more efficient polymeric matrix for MB release studies when comparing with the chitosan/pectin one. Thus, solute release processes could be efficiently performed in gastrointestinal systems without significantly losing efficiency.

Figure 9 and Figure 10 show the results of the pseudo-first- and pseudo-second-order kinetic models during the MB release from the chitosan/pectin and chitosan/DNA hydrogels in different fluids at 25 and 37 °C. Table 4 and Table 5 summarize the corresponding fitting kinetic parameters.

The best kinetic model that fits the MB release was defined by assessing the following parameters: R^2^, χ^2^ and AIC. The AIC statistical parameter is commonly employed to confirm if the experimental results fit the theoretical model. Overall, the pseudo-first-order kinetic model is the most appropriate to explain the MB release process when there is a low MB concentration inside the polymer network. On the contrary, the pseudo-second-order kinetic model demonstrated to be more appropriate when there is high MB concentration inside the polymer network [44]. At high MB concentration inside the polymer network, the occurrence of chemisorption between MB and the hydrogel, affecting desorption/release process, was observed. This effect is associated with the rearrangement of the hydrogel chains after MB interactions with active adsorption sites.

## 4. Conclusions

Physically crosslinked chitosan/pectin blend and chitosan/DNA blend hydrogels were efficiently prepared by coacervation using an emulsion (oil-in-water) technique. It has been found that, in general, pectin-containing gel shows a degree of swelling higher than those containing DNA. The exception occurs in the simulated intestinal fluid media, where the DS of chitosan–pectin, although low, is one-half that obtained for the chitosan DNA. Theses values suggest that the blends are dependent on the protonotation/deprotonation properties of pectin and DNA and the blends behave as polyelectrolytes. It was interesting to find out that the water sorption mechanism is controlled by diffusion (i.e., pseudo-Fickian) and the rate is always higher for the chitosan–pectin blend, which agrees with the higher water-free volume that characterizes this matrix. However, after MB loading, some changes in the physical–chemical properties of the hydrogels were noticed. For example, the thermal stability of the chitosan–DNA increases in 9 °C whilst a slight decrease (of 3 °C) in the maximum temperature of degradation of chitosan–pectin occurs. The modification of the polymeric structure upon MB sorption was also corroborated by the analysis of the hydrogels’ surface morphology. The sorption isotherm analysis shows that the interaction mechanism between the MB and the gels, in water media, occur via Redlich–Peterson and Sips models for pectin- and DNA-containing gels, respectively. This indicates that the sorption of MB in the former gel occurs by mono- and multilayer sorption, whilst in the latter, the data analysis suggests that the sorption occurs mainly by monolayer interaction, especially at the highest MB concentrations, where the Sips model approaches the Langmuir one. Such an effect is consistent with the previous swelling degree and water diffusion analysis as well as with the highest MB loading and encapsulation efficiencies for chitosan–pectin.

The MB release from different gels to different simulated fluids at 25 and 37 °C was also measured. The MB release follows a pseudo-second-order release kinetics showing that MB-polymer or MB-MB interactions playing an important role in the release mechanism. The cumulative release of MB is significant in the simulated gastric fluid media, reaching values of 55% and 96% for chitosan–pectin and chitosan–DNA blends, respectively, at 25 °C. These values decrease by 20% and 50%, respectively, in the case of intestinal fluid. The release of MB shows that the interaction between the polymer and MB, mainly electrostatic, is lower in the DNA-containing gel, allowing the release of almost all loaded MB. Overall, chitosan–pectin and chitosan–DNA physically crosslinked hydrogels obtained in this work could be tested for drug release in the gastrointestinal tract with high performance.

## Figures and Tables

**Figure 1 pharmaceutics-13-00842-f001:**
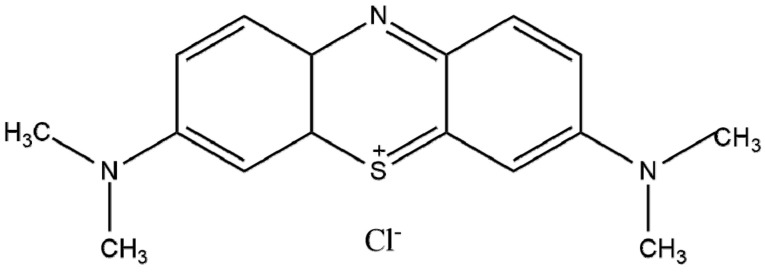
Chemical structure of methylene blue.

**Figure 2 pharmaceutics-13-00842-f002:**
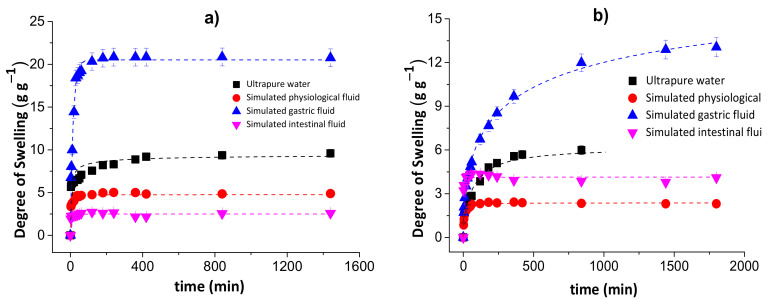
Degrees of swelling for the chitosan/pectin (**a**) and chitosan/DNA (**b**) blend hydrogels in different media.

**Figure 3 pharmaceutics-13-00842-f003:**
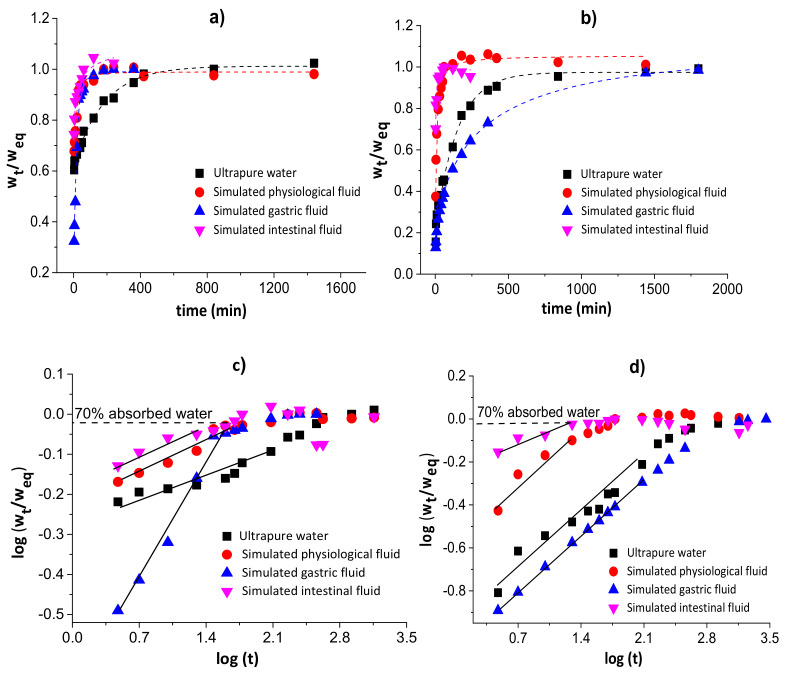
Water diffusion rates (**a**,**b**) and linear regressions for 60% of absorbed water (**c**,**d**) in chitosan/pectin (**a**,**c**) and chitosan/DNA (**b**,**d**) blend hydrogels.

**Figure 4 pharmaceutics-13-00842-f004:**
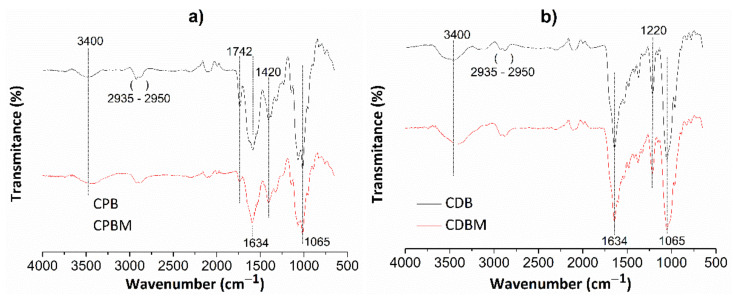
FT-IR spectra for chitosan/pectin (**a**) and chitosan/DNA (**b**) hydrogels without (CPB and CDB, respectively) and with MB (CPBM and CDBM, respectively).

**Figure 5 pharmaceutics-13-00842-f005:**
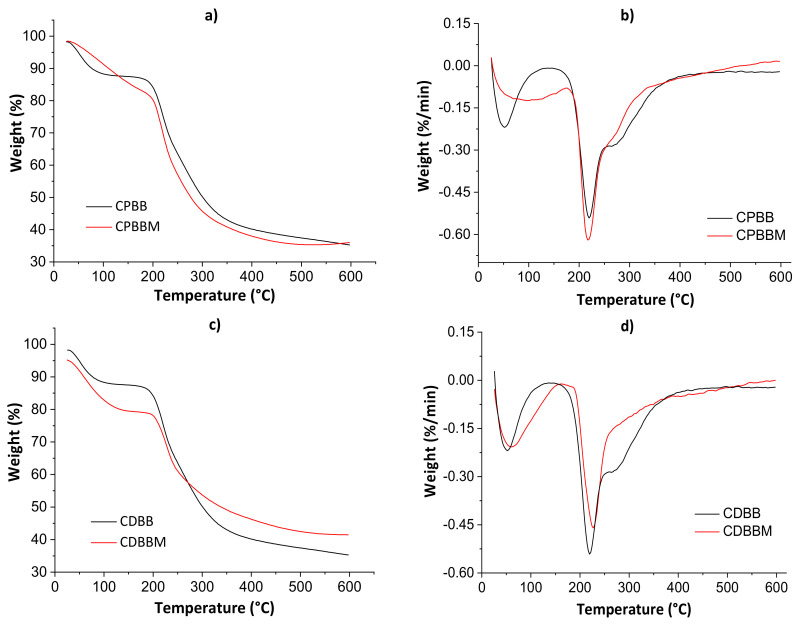
Thermogravimetric curves (**a**,**c**) and their derivatives (**b**,**d**) for both blends before and after the MB encapsulation.

**Figure 6 pharmaceutics-13-00842-f006:**
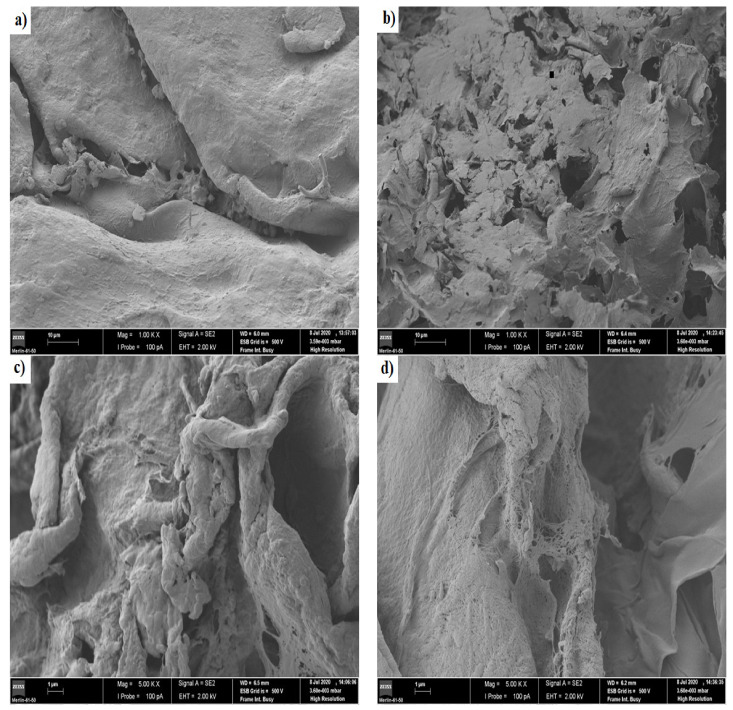
Images of scanning electron microscopy for the chitosan/pectin hydrogel (**a**,**b**) and chitosan/DNA hydrogel (**c**,**d**) before loading methylene blue (**a**,**c**) and after loading methylene blue (**b**,**d**).

**Figure 7 pharmaceutics-13-00842-f007:**
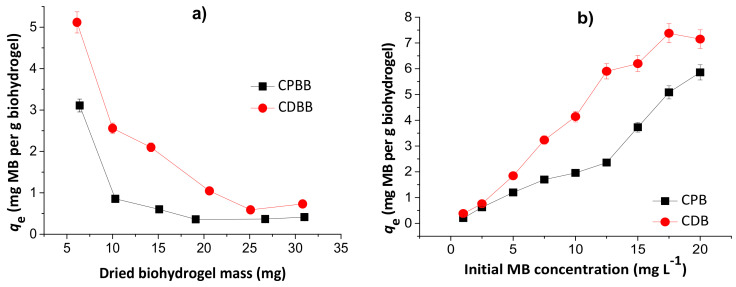
Effects of dried hydrogel mass (**a**) and initial MB concentration (**b**) during adsorption studies to chitosan/pectin blend hydrogel (CPB) and chitosan/DNA blend hydrogel (CDB). Experimental conditions: initial MB concentration of 10.0 mg L^−1^ (Figure 7a), dried hydrogel mass of 10.0 mg (Figure 7b), temperatures of 25 °C and adsorption times of 24 h.

**Figure 8 pharmaceutics-13-00842-f008:**
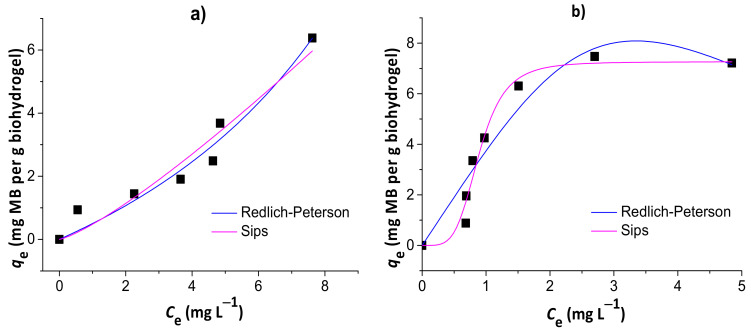
Results of nonlinear Redlich–Peterson and Sips isotherm models for the adsorption of MB to the chitosan/pectin (**a**) and chitosan/DNA (**b**) blend hydrogels.

**Figure 9 pharmaceutics-13-00842-f009:**
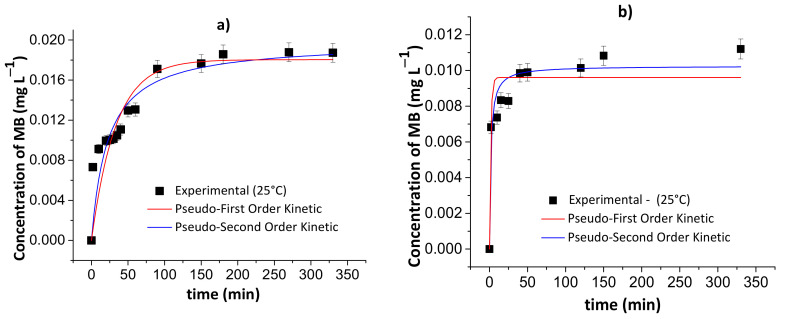
Results of the pseudo-first- and pseudo-second-order kinetic models during the controlled MB release from the chitosan/pectin blend hydrogel in simulated gastric fluid at 25 °C (**a**) and 37 °C (**c**), and simulated intestinal fluid at 25 °C (**b**) and 37 °C (**d**). Experimental conditions: dried hydrogel mass of 10.0 mg and constant stirring of 110 rpm.

**Figure 10 pharmaceutics-13-00842-f010:**
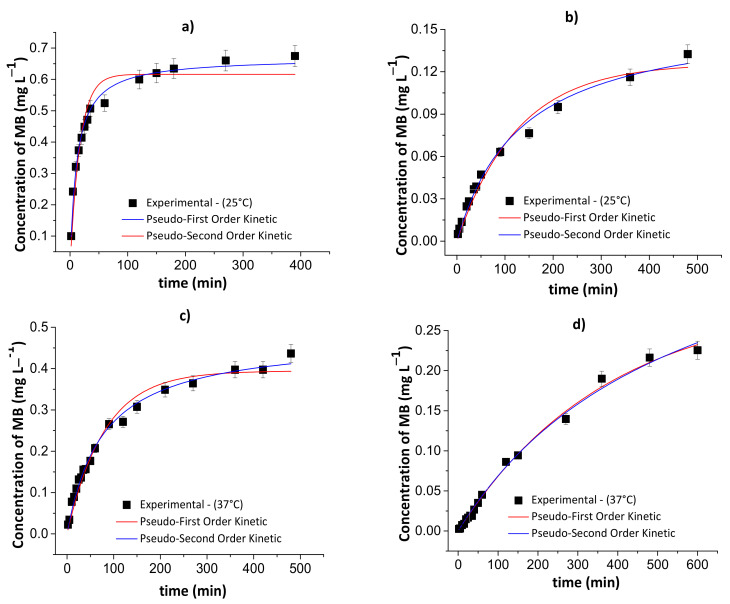
Results of the pseudo-first- and pseudo-second-order kinetic models during the controlled MB release from the chitosan/DNA blend hydrogel in simulated gastric fluid at 25 °C (**a**) and 37 °C (**c**), and simulated intestinal fluid at 25 °C (**b**) and 37 °C (**d**). Experimental conditions: dried hydrogel mass of 10.0 mg and constant stirring of 110 rpm.

**Table 1 pharmaceutics-13-00842-t001:** Swelling parameters and error analyses for the absorption of water in chitosan/pectin (CPBB) and chitosan/DNA (CDBB) blend hydrogels in different media.

Hydrogel	Diffusion Exponent (n)	*k* (min^−n^)	R²	$\chi^{2}$	AIC
	Ultrapure water				
CPBB	0.081	0.542	0.899	3.1 × 10^−4^	−26.54
CDBB	0.323	0.126	0.956	0.002	−20.36
Simulated physiological fluid					
CPBB	0.126	0.580	0.954	1.8 × 10^−4^	−17.41
CDBB	0.379	0.271	0.923	0.002	−5.761
Simulated gastric fluid					
CPBB	0.401	0.204	0.981	7.9 × 10^−4^	−16.72
CDBB	0.370	0.086	0.999	1.7 × 10^−5^	−37.74
Simulated intestinal fluid					
CPBB	0.097	0.680	0.928	1.5 × 10^−4^	−10.61
CDBB	0.143	0.617	0.921	6.8 × 10^−4^	−9.07

**Table 2 pharmaceutics-13-00842-t002:** Thermal analysis parameters for chitosan/pectin without (CPB) and with MB (CPBM) blend hydrogels, and chitosan/DNA without (CDB) and with MB (CDBM) blend hydrogels.

Stage	Parameter	CPB	CPBM	CDB	CDBM
1st stage	weight (%)	5.455	7.282	5.260	10.95
	*T*_onset_ (°C)	52	97	52	63
2nd stage	weight (%)	13.17	18.03	12.62	21.04
	*T*_onset_ (°C)	181	190	169	186
	*T*_max_ (°C)	220	217	221	229
Final	Residue (%)	35.27	35.99	35.27	41.47

**Table 3 pharmaceutics-13-00842-t003:** Parameters of nonlinear Langmuir, Freundlich, Redlich–Peterson and Sips isotherm models for the adsorption of MB to the chitosan/pectin (CPB) and chitosan/DNA (CDB) blend hydrogels.

Langmuir Isotherm						
Hydrogel	*K*_L_ (L mg^−1^)	*q*_max_ (mg g^−1^)	*R* _L_	R²	χ^2^	AIC
CPBB	5.3 × 10^−6^	116.9	0.999	0.892	0.478	2.73
CDBB	0.480	11.60	0.455	0.822	1.480	6.35
Freundlich Isotherm						
	*K*_F_ (mg^(n−1)/n^ L^1/n^g^−1^)	*n*	*b* _F_	R²	χ^2^	AIC
CPBB	0.343	0.698	1.432	0.940	0.268	0.97
CDBB	3.657	1.912	0.523	0.755	0.956	7.47
Redlich-Peterson Isotherm						
	α_R-P_ (mg^−1^)	*K*_R-P_ (L g^−1^)	*β* _R-P_	R²	χ^2^	AIC
CPBB	0.061	0.470	0.969	0.951	0.215	0.30
CDBB	0.022	3.823	0.121	0.885	0.956	4.21
Sips Isotherm						
	*β* _S_	*K*_S_ (L mg^−1^)	*q*_ms_ (mg g^−1^)	R²	χ^2^	AIC
CPBB	1.256	0.003	188.6	0.941	0.263	−1.46
CDBB	4.398	1.648	7.264	0.967	0.271	−0.18

**Table 4 pharmaceutics-13-00842-t004:** Fitting parameters of the pseudo-first- and pseudo-second-order kinetic models for the release of MB from the chitosan/pectin blend hydrogel in different fluids.

Model	*k*_1_ (min^−1^)	*k*_2_ (L mg^−1^min^−1^)	R²	χ²	AIC
Simulated gastric fluid (25 °C)					
Pseudo-first-order	0.0301	-	0.793	5.64 × 10^−6^	−73.66
Pseudo-second-order	-	2.267	0.848	4.14 × 10^−6^	−75.68
	Simulated intestinal fluid (25 °C)				
Pseudo-first-order	0.604	-	0.820	2.02 × 10^−6^	−51.74
Pseudo-second-order	-	57.769	0.889	1.24 × 10^−6^	−54.02
	Simulated gastric fluid (37 °C)				
Pseudo-first-order	0.024	-	0.891	2.75 × 10^−6^	−67.24
Pseudo-second-order	-	1.818	0.907	2.34 × 10^−6^	−68.15
	Simulated intestinal fluid (37 °C)				
Pseudo-first-order	0.067	-	0.881	1.41 × 10^−6^	−59.32
Pseudo-second-order	-	10.919	0.949	5.98 × 10^−7^	−63.42

**Table 5 pharmaceutics-13-00842-t005:** Fitting parameters of the pseudo-first- and pseudo-second-order kinetic models for the release of MB from the chitosan/DNA blend hydrogel in different fluids.

Model	*k*_1_ (min^−1^)	*k*_2_ (L mg^−1^min^−1^)	R²	χ²	AIC
Simulated gastric fluid (25 °C)					
Pseudo-first-order	0.059	-	0.927	0.002	−32.40
Pseudo-second-order	-	0.129	0.987	3.76 × 10^−4^	−42.87
Simulated intestinal fluid (25 °C)					
Pseudo-first-order	0.008	-	0.975	4.28 × 10^−5^	−51.74
Pseudo-second-order	-	0.047	0.989	1.82 × 10^−5^	−56.55
Simulated gastric fluid (37 °C)					
Pseudo-first-order	0.013	-	0.973	4.58 × 10^−4^	−58.36
Pseudo-second-order	-	0.028	0.991	1.52 × 10^−4^	−67.48
Simulated intestinal fluid (37 °C)					
Pseudo-first-order	0.003	-	0.995	2.78 × 10^−5^	−72.385
Pseudo-second-order	-	0.004	0.995	3.35 × 10^−5^	−71.00

## Data Availability

The data presented in this study are available in this article.

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
