# Peer review of "Methylene Blue Release from Chitosan/Pectin and Chitosan/DNA Blend Hydrogels"

_pharmaceutics, 2021, doi:10.3390/pharmaceutics13060842_

Round 1

Reviewer 1 Report

I believe the manuscript by Cesco et al. is of great interest, as the field of hydrogels used for the release of bioactive molecules is continuously growing. 

However, before publication, the manuscript should undergo an extensive editing of English, as there are many gramatical errors that make it difficult to follow.

Additionally, there are a few minor revisions that should be addressed:

The introduction should contain more background information regarding previous uses of chitosan hydrogels containing dyes.

Lines 151-159: The information should be summarized in a Table because it is difficult to follow in the current state.

I do not understand the difference between section 2.4.1. and 2.5 regarding the loading of MB within the hydrogels. Please clarify.

Several references should be introduced:

https://doi.org/10.33263/BRIAC101.706713

https://doi.org/10.1016/j.reactfunctpolym.2020.104699

https://doi.org/10.33263/BRIAC101.803810

Reviewer 2 Report

This manuscript is concerned with the preparation of hybrid biohydrogels for the slow and the controlled release of molecules such as methylene blue.

Though the manuscript holds extensive experimental physical characterization work yet it lack novelty and the idea has been introduced previously and with the same materials in several studies:

Budi Hastuti and Saptono Hadi 2019 IOP Conf. Ser.: Mater. Sci. Eng. 617 012002

RSC Adv., 2018,8, 14609-14622

Polymers 2019, 11, 1837; doi:10.3390/polym11111837

https://www.sciencedirect.com/science/article/abs/pii/S0141813016307838#

Colloid Journal volume 82pages311–323(2020)

Biomacromolecules May-Jun 2004;5(3):928-36. doi: 10.1021/bm034502r.

Reviewer 3 Report

In general, the manuscript has to be corrected and more correct scientific vocabulary has to be used. 

Line 56, in this work were not studied "proteins and aminoacids" please correct. Moreover, when you have mentioned in line 58 that they are inexpensive,,,really? pure proteins are quite expensive.

Line 62 eliminate "and so forth" it is not scientific.

Line 82 for reproducibility, the MW of chitosan has to be added to the Materials.

Line 83 are the authors sure that the percentage of pectin was esterification and not % of galacturonic acid. Also, the Mw of this chemical has to be added otherwise the value of viscosity to identify the polysaccharide. 

Line 110. why the authors studied the kinetics in gastric fluid. It could be interesting that the authors explain the administration of MB to the body. 

Line 237, what is the meaning of biohydrogel.

Line  474, the definition of coacervation does not imply "chemical modification" please correct with a more appropriate verb than synthesize

In general, it is important that the authors clarify in the conclusions the main idea to prepare two kinds of hydrogels, advantages or disadvantages between them, because a conclusion "both" they can be used in very strange. It could be also necessary to add cytotoxicity of the materials to identify the best candidate. Otherwise, a good evaluation of the physiochemical properties of the hydrogels has to be done. Please, correct the conclusions.

Round 2

Reviewer 2 Report

The article still lacks novelty especially with the use of methylene blue as a model drug. I do not think the article is suitable to be publishes in a highly ranked journal such as Pharmaceutics.

Author Response

We respectfully disagree with Reviewer#2. Unfortunately, we did not receive any grounded response to our previous reply but instead a subjective statement.

Reviewer 3 Report

The authors have improved the weak conclusions, which for me was a poor part. Now, I recommend this manuscript for publication

Author Response

Thank you for your comment and recommendation.